# Benchmarking of novel green fluorescent proteins for the quantification of protein oligomerization in living cells

**Annett Petrich[1], Amit Koikkarah Aji[1], Valentin Dunsing[1,2], Salvatore Chiantia[1] ***

**1** University of Potsdam, Institute of Biochemistry and Biology, Potsdam, Germany, **2** Aix-Marseille University, CNRS, UMR 7288, IBDM, Turing Center for Living Systems, Marseille, France

* chiantia@uni-potsdam.de

## Abstract

Protein-protein-interactions play an important role in many cellular functions. Quantitative non-invasive techniques are applied in living cells to evaluate such interactions, thereby providing a broader understanding of complex biological processes. Fluorescence fluctuation spectroscopy describes a group of quantitative microscopy approaches for the characterization of molecular interactions at single cell resolution. Through the obtained molecular brightness, it is possible to determine the oligomeric state of proteins. This is usually achieved by fusing fluorescent proteins (FPs) to the protein of interest. Recently, the number of novel green FPs has increased, with consequent improvements to the quality of fluctuation-based measurements. The photophysical behavior of FPs is influenced by multiple factors (including photobleaching, protonation-induced "blinking" and long-lived dark states). Assessing these factors is critical for selecting the appropriate fluorescent tag for live cell imaging applications. In this work, we focus on novel green FPs that are extensively used in live cell imaging. A systematic performance comparison of several green FPs in living cells under different pH conditions using Number & Brightness (N&B) analysis and scanning fluorescence correlation spectroscopy was performed. Our results show that the new FP Gamillus exhibits higher brightness at the cost of lower photostability and fluorescence probability (*pf*), especially at lower pH. mGreenLantern, on the other hand, thanks to a very high *pf*, is best suited for multimerization quantification at neutral pH. At lower pH, mEGFP remains apparently the best choice for multimerization investigation. These guidelines provide the information needed to plan quantitative fluorescence microscopy involving these FPs, both for general imaging or for protein-protein-interactions quantification via fluorescence fluctuation-based methods.

## Introduction

A multitude of cellular processes, such as biomolecule transport, ion channel activity, cell-cell adhesion and communication are regulated by protein-protein-interactions (PPIs) [1–3]. "Classical" bulk biochemical *in vitro* methods that are used to quantify PPIs (e.g., co-

**Data Availability Statement:** All plot data are included in this online submission as Excel file.

**Funding:** Deutsche Forschungsgemeinschaft (DFG) project number 407961559 to S.C. www.

dfg.de HFSP long-term postdoctoral fellowship (HFSP LT0058/2022-L) to V.D. www.hsfp.org The funders had no role in study design, data collection and analysis, decision to publish, or preparation of the manuscript.

**Competing interests:** The authors have declared that no competing interests exist.

**Abbreviations:** FFS, fluorescence fluctuation spectroscopy; FP, fluorescent protein; N&B, number and brightness; pf, fluorescence probability; PM, plasma membrane; PPI, protein-protein-interaction; sFCS, scanning fluorescence correlation spectroscopy.

immunoprecipitation (co-IP), pull-down assays and western blotting) cannot be used to obtain information about intracellular protein distribution in live-cell samples or to monitor the effects of variations in concentrations between different cells [4, 5]. Conventional optical microscopy can visualize the localization of proteins, but its resolution is limited [4, 6]. More complex approaches, such as fluorescence fluctuation spectroscopy (FFS), can assess the interactions between molecules in complexes and obtain insights into cellular pathways and assembly processes [4–8]. FFS provides information about dynamics through the analysis of signal fluctuations from fluorescently labeled molecules [6, 7, 9]. Additionally, the magnitude of such fluctuations can be used to derive quantitative information about the multimerization state (i.e., number of monomers in a multimer) of the protein of interest [7–10].

A common strategy to investigate PPIs *in cellula* via FFS is the fusion of a fluorescent protein (FP) to the protein of interest [7–9, 11]. By comparing the brightness of protein multimers tagged with FPs to the brightness of a monomeric reference, it is possible to quantify the number of FP monomers in the complex and, thus, the oligomerization state of the protein of interest [5, 7, 12]. A major problem for several FFS applications that rely on FPs, though, is the presence of non-emitting "dark" proteins, which can be quantified through the so-called fluorescence probability (*pf*) [7, 13]. It is sometime assumed that FPs have a *pf* value of 1 meaning that, for example, a trimer containing three FPs emits in average a three-fold higher signal than a monomer labelled with one FP [5, 7]. Instead, *pf* values of e.g. green emitting FPs have been reported in the range between 0.5 and 0.8 [1, 2, 7, 13–19]. Recently, we have systematically quantified the *pf* of several FPs, focusing mainly on proteins emitting in the red part of the visible spectrum [7]. In summary, since high *pf* values are required for increased sensitivity, not all FPs are equally suitable for oligomerization studies [7]. Because of the presence of non-emitting FPs (i.e., *pf* lower than 1), FFS approaches might underestimate the amount of FPs and therefore, the oligomeric state of the protein of interest.

Some of the most used FPs are the green fluorescent protein (GFP) and its mutants, such as the monomeric enhanced GFP (mEGFP) [5, 20]. This FP has a high quantum yield (QY), enhanced photostability, and minimal interference with the cellular machinery [20]. Of note, several variants have been engineered over the past few years (e.g. mNeonGreen (mNG), mGreenLantern (mGL), and Gamillus) in an effort to optimize molecular brightness, folding efficiency, photostability, and pH stability of the fluorescent probe (Table 1) [21–24].

In the context of complex experiments in living cells, a FP might encounter different environments and pH conditions (e.g. acidic pH in lysosomes, secretory granule and endosomes; neutral pH in the nucleus, cytoplasm and endoplasmic reticulum (ER); basic pH in mitochondria and a pH gradient in the Golgi network) [22, 25]. Measurements of GFP at acidic pH have shown a decrease of the *pf* value, possibly due to an increase of proteins in the dark-state [22, 26–28]. Consequently, it would be useful to systematically analyze the properties of the above-

**Table 1. Photophysical characteristics of the fluorescence proteins (FPs).** QY, Quantum yield; norm., normalized.

| FP | $\lambda_{ex}$ [nm] | $\lambda_{em}$ [nm] | QY | Brightness* | cellular norm. <I> | pK$_a$ | maturation [min] | Reference |
|---|---|---|---|---|---|---|---|---|
| mEGFP | 488 | 507 | 0.71 | 38.0 | 1.0[#,†] | 6.0 | 28.0 | [21] |
| mNG | 506 | 516 | 0.78 | 90.0 | 3.3[#] | 5.7 | 18.0 | [21] |
| mGL | 503 | 514 | 0.72 | 74.0 | 6.3[#] | 5.6 | 14.0 | [21] |
| Gamillus | 504 | 519 | 0.90 | 74.7 | 0.5[†] | 3.4 | 8.0 | [24] |

* (extinction coefficient × QY)/1,000, where QY and extinction coefficients are measured at the absorbance peak ($\lambda_{ex} \pm 2$ nm)

[#] Average intensity in FP-expressing BE (2)-M17 human neuroblastoma cells, relative to EGFP, using the P2A quantitative co-expression system

[†] Average intensity in FP-expressing HeLa cells, relative to EGFP

mentioned novel FPs, in order to perform an exact quantification of PPIs, also in different intra-cellular environments.

Here, we benchmarked the performance of novel green FPs (mGL and Gamillus) against the well-established mEGFP and mNG. The presence of non-fluorescent states and photo-stability under different pH conditions were measured using Number & Brightness (N&B) and scanning fluorescence correlation spectroscopy (sFCS) analysis [5]. Our results indicate that some of the observed proteins are brighter, although unstable under longer/stronger illumination. Furthermore, we identify which proteins are more suitable for multimerization quantification, rather than "simple" imaging, also at different pH conditions.

## Materials and methods

### Fluorescent protein constructs

A description of the cloning procedure for all constructs can be found in the Supporting Information. All plasmids generated for this work will be available on Addgene (https://www.addgene.org/). A schematic overview of the intracellular localization of the constructs is provided in S1A Fig in S1 File and an overview of the linker sequences within the FP structures is provided in S1 Table in S1 File.

### Cell culture

Human embryonic kidney cells from the 293T line (HEK293T, CRL-3216[TM]) and Chinese hamster ovary cells (CHO-K1, CCL-61™) were purchased from ATCC (Kielpin Lomianki, Poland). Both cell lines were maintained in phenol red-free Dulbecco's modified Eagle's medium (DMEM) supplemented with 10% fetal bovine serum, 2 mM L-glutamine, 100 U/mL penicillin, and 100 μg/mL streptomycin in a humidified incubator with 5% $CO_2$ at 37˚C. Cells were passaged every 2–3 days after reaching ca. 80% confluence in tissue culture flask, for a maximum of 15 passages. All solutions, buffers, and media used for cell culture were purchased from PAN-Biotech (Aidenbach, Germany).

### Preparation for microscopy experiments

For microscopy, $6 \times 10^5$ (HEK293T) or $4 \times 10^5$ (CHO-K1) cells were plated in 35 mm dishes (CellVis, Mountain View, CA, USA) with optical glass bottom (#1.5 glass, 0.16–0.19 mm), 24 h before transfection. HEK293T cells are more suitable for sFCS measurements since they are relatively thick and therefore preferable for sFCS acquisition perpendicular to the PM. CHO-K1 cells are rather flat and therefore more suitable for N&B measurements in the cytoplasm. Cells (40–50% confluency) were transfected 16–24 h before imaging using 200 ng plasmid DNA per dish with Turbofect (Thermo Fisher Scientific, Waltham, MA, USA) according to the manufacturer's instructions. For measurements under different pH conditions, the culture medium was exchanged with buffer containing 140 mM NaCl, 2.5 mM KCl, 1.8 mM $CaCl_2$, 1.0 mM $MgCl_2$, and 20 mM HEPES with a pH value of 5.6, 7.4 or 9.2, and incubated for 5 minutes.

### Confocal microscopy system and setup calibration for fluorescence fluctuation spectroscopy

All measurements were performed on a Zeiss LSM780 microscope (Carl Zeiss Microscopy GmbH, Oberkochen, Germany) using a Plan-Apochromat 40×/1.2 Korr DIC M27 water immersion objective and a 32-channel GaAsP detector. FPs were excited with a 488 nm Argon laser (488 nm dichroic mirror) and the fluorescence signal was collected in the range of 498 to

606 nm in photon-counting mode. For the spectral analysis under different pH conditions, fluorescence was detected in spectral channels of 8.9 nm width (23 channels between 491 nm and 690 nm). To decrease out-of-focus light, a one airy unit pinhole was used. All measurements were performed at $22 \pm 1°C$. Cells were incubated for 5 min at room temperature before each measurement.

The confocal volume was calibrated daily by performing point FCS measurements with Alexa Fluor® 488 (AF488, Thermo Fischer, Waltham, MA, USA) in water at 30 nM, with the same laser power and beam path used for N&B and sFCS measurements. Prior to that, the signal was maximized adjusting the collar ring of the objective and the pinhole position. Then, five measurements were performed, each consisting of 15 acquisitions of 10 s, and the data was fitted using a three-dimensional diffusion model including a triplet contribution. The structure parameter $S$ (i.e., the ratio between the vertical and lateral dimension of the confocal volume) was typically around 5 to 9, and the diffusion time $\tau_d$ around 30 to 35 μs.

## N&B measurements

N&B analysis was performed as previously described [7, 10] with few modifications: 100 frames were acquired as 128x 128 pixel images with pixel dimensions of 400 nm and pixel dwell time of 50 μs. The time-stacks were analyzed using a custom MATLAB code (The Math-Works, Natick, MA, USA). The MATLAB algorithm calculates the molecular brightness and number as a function of pixel position, as described by Digman *et al.* [11]. Bleaching and minor cell movements are partially corrected using a boxcar-filter with an 8-frame window applied pixel-wise, as previously described [7, 29, 30]. Final brightness values were calculated by extrapolating the partial brightness values (i.e., calculated within each 8-frame window) to the earliest time point. Detector saturation was avoided by excluding pixels with photon-counting rates exceeding 1 MHz. A schematic overview of the N&B analysis is provided in S1C Fig in S1 File.

## sFCS measurements

Scanning FCS experiments were performed as previously described [7, 31]. Briefly, a line scan of $256 \times 1$ pixels (pixel size $\approx$80 nm) was performed perpendicular to the membrane, using a 472.73 μs scan time. 400,000 lines were acquired (total scan time $\approx$3 min) in photon counting mode. Laser power values were typically between $\approx$1.5 μW (brightness analysis) and $\approx$6 μW (photobleaching analysis). Measurements were exported as TIFF files and analyzed in MATLAB (The MathWorks, Natick, MA, USA) using a custom-written script as previously described [7, 31, 32]. The obtained autocorrelation curves $G(\tau)$ were analyzed using a two-dimensional Brownian diffusion model [33]: $\frac{1}{N} \left( 1 + \frac{\tau}{\tau_d} \right)^{-1/2} \left( 1 + \frac{\tau}{\tau_d S^2} \right)^{-1/2}$, where $S$ is the structure parameter, $\tau_d$ is the diffusion time and $N$ is the amount of fluorescent particles in the detection volume. The average molecular brightness is then calculated as the ratio between the average fluorescence intensity $<I>$ and the particle number $N$. A schematic overview of the sFCS analysis is provided in S1D Fig in S1 File.

## Brightness calibration and fluorophore maturation

The molecular brightness, i.e. the average amount of photons emitted by a molecule in the unit of time, is directly connected to the oligomeric state of protein complexes. This is based on the typical premise that all fluorophores within an oligomer are fluorescent. However, FPs can be in a non-mature state, undergo dark state transitions or, in general, be non-fluorescent [8]. We characterize all the processes leading to non-fluorescent FPs using a single parameter, the

apparent fluorescence probability (*pf*), i.e. the probability of a FP to emit a fluorescence signal. The determination of the *pf* value from apparent brightness values was performed as previously described [7, 32]. Shortly, for each sample and each experiment day, the average brightness for a monomeric construct was determined from multiple cells (see e.g. Figs 2A and 3A). Then, measurements were performed in several cells expressing the dimer constructs (S2 Fig in S1 File) and, from each of these measurements and the average monomer brightness obtained before, one *pf* value was calculated using the formula $pf = Brightness_{dimer}/Brightness_{monomer} - 1$ [7, 32]. The final *pf* value was calculated as mean of such a set of measurements (see Figs 2B and 3B).

## Statistical analysis

Results from at least three independent measurements were pooled and visualized using GraphPad Prism ver. 9.0.0 (GraphPad Software, LCC, San Diego, CA, USA). All results are displayed as box plots with each point corresponding to a measurement in a single cell or as mean ± standard error of the mean (SEM) plots. Median values and whiskers indicating minimum and maximum values are displayed in the box plots. The mean, median, interquartile range (IQR) are indicated in each graph together with the sample size. Significance values are given in each graph and figure captions, respectively. Statistical significance was tested by using D'Agostino-Pearson normality test followed by one-way ANOVA analysis and the Tukey's or Fishers Least Significant Difference (LSD) multiple comparisons test.

# Results

## mGL and Gamillus are less photostable than mEGFP or mNG

All monomeric FPs were evaluated for photostability at different laser powers using confocal microscopy, in the context of a typical N&B experiment performed in CHO-K1 cells. Such an approach simply consists of a time-series acquisition with continuous scanning over ca. two minutes. mGL and Gamillus displayed higher photobleaching, both in the cytosol and at the PM, in comparison to mEGFP and mNG (Fig 1A and 1B, S2 Table in S1 File). For example, ≈30% of the total mGL was bleached after a ca. two-minute of continuous scanning, using 2.4 µW excitation power. At the same time, N&B analysis provided the brightness of each FP, as a function of the laser power. Although an accurate analysis should be performed at an excitation power causing as low as possible bleaching (see next paragraph), Fig 1C and 1D confirm that the observed brightness predictably increases roughly linearly with the laser power. As expected, due to the detection geometry, a higher signal is observed for molecules restricted to the PM [34]. Finally, Gamillus shows generally a higher brightness, as especially noticeable for measurements at the PM or at higher excitation powers.

## Gamillus exhibits high brightness but low fluorescence probability

To compare the brightness effectively observed in a typical confocal microscopy setup and, specifically, in the context of FFS experiments, we have performed in-depth N&B analysis of FP monomer and dimer constructs. In order to maintain bleaching below ≈15%, we have excited the fluorophores with a laser power of 1.2 µW. CHO-K1 cells were transfected with the required plasmids and observed after 16 h. Fig 2A shows the brightness values measured for the different FP monomer constructs, either in the cytosol or at the PM. The results follow, as expected, the general trend described above (Fig 1C and 1D). No significant difference can be observed between mean brightness values at this low laser power, with the exception of mp-Gamillus displaying a higher mean brightness than the other membrane-associated constructs.

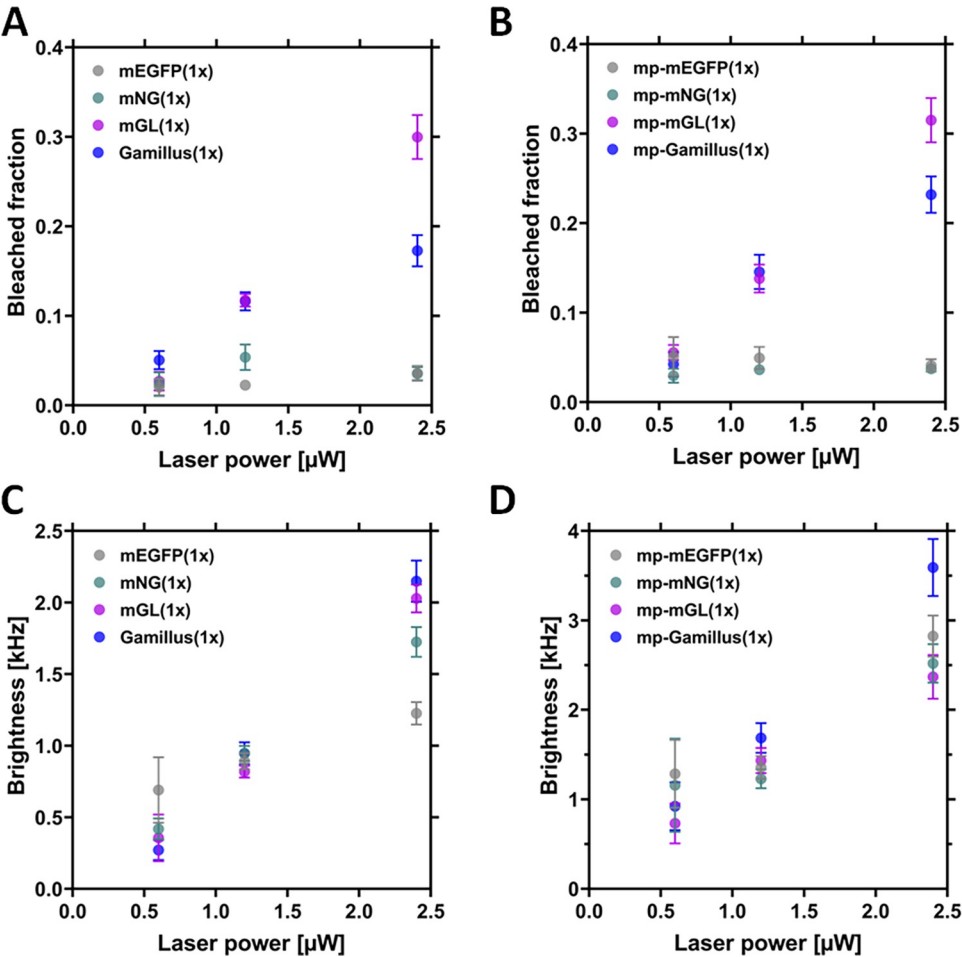

**Fig 1. Comparison of the molecular brightness and bleached fraction for the monomeric FPs, as obtained from N&B measurements in CHO-K1 cells.** CHO-K1 cells were transfected with plasmids coding for the monomeric cytosolic FPs (mEGFP(1x), mNeonGreen(1x) (here, called mNG(1x)), mGreenLantern(1x) (here called mGL(1x)) and Gamillus(1x)) or with membrane-anchored FPs. The latter are anchored to the inner leaflet of the PM via a myristoylated and palmitoylated (mp) peptide (mp-mEGFP(1x), mp-mNG(1x), mp-mGL(1x) or mp-Gamillus(1x)). N&B measurements were performed ≈16 h after transfection with a laser power of 0.6 µW, 1.2 µW and 2.4 µW. (A-B): Mean bleached fractions measured for cytosolic FPs (A) and for PM-anchored FPs (B), as a function of laser power. The bleached fraction indicates the amount of fluorescence signal lost after a N&B measurement. Error bars indicate the standard error of the mean (SEM). (C-D): Mean brightness values measured for cytosolic FPs (C) and for PM-anchored FPs (D), as a function of laser power. Error bars represent the SEM. Exact values and sample sizes are summarized in S2 Table in S1 File.

Next, we have performed similar experiments on dimer constructs of the same FPs (S2A Fig and S3 Table in S1 File), in order to calculate the *pf*, as shown in Fig 2B. All *pf* values are in the expected range, around 0.7, and no large difference can be observed between the different fluorophores in general. Noticeably though, mp-Gamillus exhibits *pf* values lower than those of the other FPs (in particular, significantly lower than mp-mGL, ca. 0.6 vs. ca. 0.9).

## Performance comparison of FPs at different pH values

The fluorescence emission intensity of some FPs often decreases at low pH values due to protonation of the chromophore [22, 24, 26]. Protonation can induce the transition to dark states and, therefore, negatively affect the accuracy of molecular brightness measurements [28].

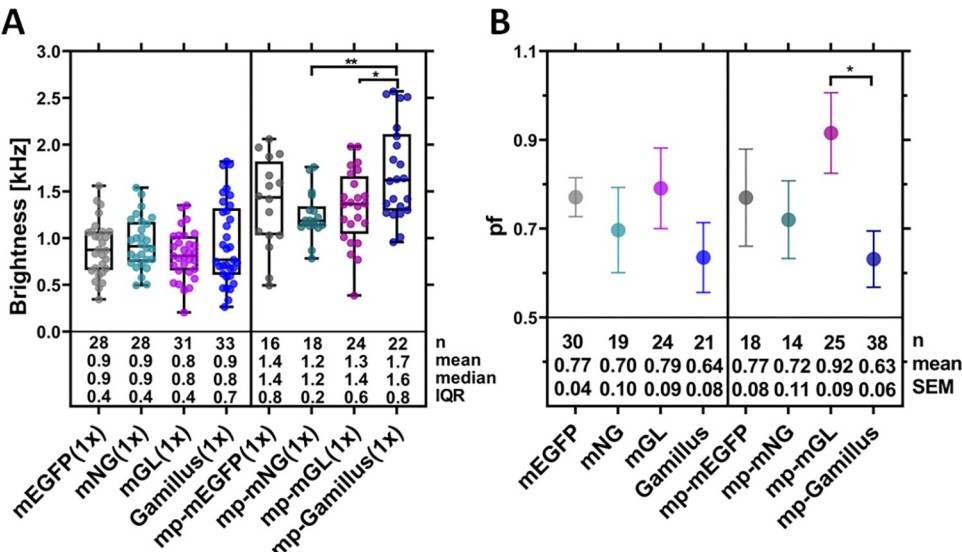

**Fig 2. Comparison of brightness and fluorescence probability (*pf*) for the examined green FPs, as obtained from N&B measurements in CHO-K1 cells.** N&B measurements were performed ≈16 h after transfection, using a laser power of 1.2 μW. (A): Box plot of the molecular brightness for the examined cytosolic and membrane-anchored FPs (i.e., mp-FP). Each point represents the average value measured in a single cell, from three independent experiments. Median values and whiskers ranging from minimum to maximum values are displayed. (B): Mean *pf* values calculated for the cytosolic and membrane-anchored FPs, using the brightness values measured for the corresponding FP dimers (S2A Fig in S1 File). Data were collected from three independent experiments. The error bars represent the SEM. Sample size, mean/median, and interquartile range (IQR)/SEM are indicated below the graph. Statistical significance within both plots for selected sample pairs were determined using one-way ANOVA Tukey´s multiple comparison test (* p < 0.05, ** p < 0.005).

Here, the pH-dependent fluorescence of the FPs was analyzed via confocal microscopy in the pH range from 5.6 to 9.2 in HEK293T cells using constructs associated to the outer side of the PM via a glycosylphosphatidylinositol (GPI) anchor. The fluorescence emission spectra of the different FPs did not change considerably in the assessed pH range and no photoconversion was observed (S3 Fig in S1 File). Also, the photostability of the FP variants at different pH conditions was qualitatively compared (S4 Fig in S1 File) using an excitation power of 6 μW, on a specific position of the PM, for ca. three minutes. This configuration is usually employed for FCS measurements at the PM and is different from the N&B whole-frame scanning approach used for the experiments described in the previous paragraphs [10]. GPI-mEGFP showed a good photostability at all pH conditions, with the fluorescence signal decreasing by only ca. 20% (S4A and S5 Figs in S1 File). GPI-mNG(1x) exhibited fast initial photobleaching, particularly at pH values 5.6 and 9.2, down to ca. 50% of the original signal. At neutral pH, bleaching of GPI-mNG was only slightly higher than that of GPI-mEGFP (S4B and S5 Figs in S1 File). A fast and substantial photobleaching, especially at neutral and low pH was observed for GPI-mGL (S4C and S5 Figs in S1 File). Finally, as shown in S4D and S5 Figs in S1 File, a strong pH-dependency of photostability was observed for GPI-Gamillus: at the highest pH the bleaching was minor (i.e., ca. 20%, similar to GPI-mEGFP), while at neutral and low pH the emission signal dramatically decreased (i.e., to ca. 40% and 10% at pH 7.4 and 5.6, respectively). Overall, mEGFP showed the highest photostability for all pH conditions followed by mNG and then Gamillus (pH < 9.2) and mGL, in agreement with the results from N&B analysis (Fig 1B).

Additionally, the molecular brightness of the different FPs under various pH conditions was evaluated for monomer and dimer membrane-associated GPI-anchored constructs, and the corresponding *pf* values were determined (Fig 3, S2B Fig and S4 Table in S1 File). To

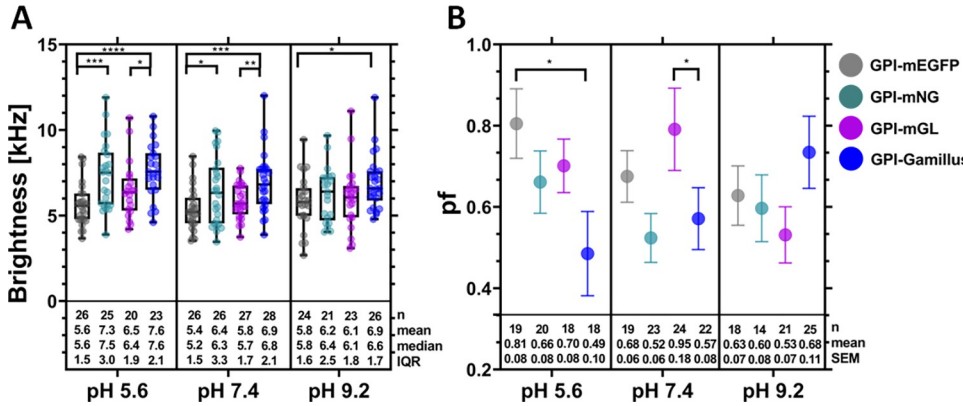

**Fig 3. Comparison of brightness and *pf* values for the examined FPs, as obtained from scanning fluorescence correlation spectroscopy (sFCS) measurements in HEK293T cells, under different pH conditions (pH 5.6, pH 7.4, and pH 9.2).** sFCS measurements were performed ≈16 h after transfection, using a laser power of 1.5 μW. (A): Box plots of the molecular brightness for the membrane-anchored FPs GPI-mEGFP, GPI-mNG, GPI-mGL and GPI-Gamillus, for three different pH conditions. Each point represents the average value measured in single cells, from three independent experiments. Median values and whiskers ranging from minimum to maximum values are displayed. (B): Average *pf* values calculated using the brightness values measured for the corresponding FP dimers (S2B Fig in S1 File), for three different pH conditions. Data were collected from three independent experiments. The error bars represent the SEM. Statistical significance was determined for selected sample pairs in both plots using one-way ANOVA Fishers Least Significant Difference (LSD) multiple comparison test (* p < 0.05, ** p < 0.005, *** p < 0.0005, **** p < 0.0001).

complement the previous experiments, we used an alternative FFS approach (i.e., sFCS [33]) and an alternative live-cell model (i.e., HEK293T) for this set of experiments. sFCS experiments result in fluorescence autocorrelation curves which can be analyzed to extract parameters such as diffusion coefficients of the labelled proteins, concentration and molecular brightness (see S6 Fig in S1 File for representative examples). As shown in Fig 3A, the results obtained at pH 7.4 corroborate what was observed also via N&B (Figs 1D and 2A), i.e. that Gamillus overall exhibits a higher brightness compared to the other FPs. This observation holds true for all the tested pH conditions. In general, the observed brightness values did not differ much among each other (being all within a ca. 15% variation interval) and we did not observe an effect of pH, at least at the low laser powers used here.

More marked differences were observed, however, for the *pf* values. In general, this parameter showed a negative correlation with increasing pH for GPI-mEGFP and a positive correlation for GPI-Gamillus. No strong correlation was observed instead for GPI-mGL or GPI-mNG. At pH 5.6, the *pf* of GPI-Gamillus was significantly lower than that of GPI-mEGFP and, overall, the lowest among all investigated proteins. At neutral pH, GPI-mGL showed the best performance, in agreement with what we observed for mp-mGL via N&B (Fig 2B). At pH 9.2, no strong differences were observed in general among all FPs, although GPI-Gamillus displayed by trend the highest *pf* value. Notably, we did not observe pH-induced changes in the autocorrelation functions (indicating e.g. significant alterations in the triplet state fraction) for any FP at this low excitation power (S6 Fig in S1 File). An in-depth study of fluorescence intensity dynamics faster than ca. 1 ms would require, in general, approaches with higher temporal resolution.

## Discussion

PPIs and, specifically, homo-multimerization can be quantified directly in living cells using quantitative fluorescence microscopy approaches based on FFS (e.g., sFCS and N&B) [6–8].

The molecular brightness (i.e., photon count rate per molecule) derived from such experiments allows the determination of the oligomeric state of FP-tagged protein complexes [7, 8, 10]. The reliability of such approaches is influenced by the photophysical behavior of the FPs, which can be summed up by the *pf* parameter [7]. The *pf* is determined by the specific experimental conditions (e.g. excitation wavelength and power, duration of illumination, geometry of the detection volume), sample environment (pH, ion composition, etc.), intracellular biochemical processes (e.g. maturation time and folding efficiency) and photophysical processes (photobleaching, quantum yield, triplet state formation, protonation- or light-induced "blinking", long-lived dark states, etc.) [9, 20, 26–28, 33, 35–40]. Therefore, three requisites should be ideally met by FPs used in typical FFS studies: i) high molecular brightness, required to achieve sufficient signal-to-noise ratio to detect single molecule fluctuations, ii) high photostability, essential to allow prolonged imaging and iii) a high *pf*, crucial for a large dynamic range and accuracy of oligomerization measurements. In this study, we present a systematic analysis of novel GFP variants (mGL and Gamillus) and discuss their suitability in the context of brightness-based oligomerization studies. Both proteins were reported to mature remarkably quickly and to possess a high brightness and medium-low acid sensitivity compared to mEGFP (see Table 1) [21, 24], thus suggesting that they are promising candidates for quantitative FFS studies. These features are, at least partially, shared also by the established GFP variant mNG [23], which is therefore also included in this study.

Previous studies have characterized these FPs and compared them to e.g. mEGFP as standard (see e.g. [41]). Nevertheless, it must be noted that essential parameters, such as brightness, pH sensitivity or photostability, are often measured with different methods and the outcome might depend on the specific setup [23, 24, 41–43]. Thus, it is reasonable to compare the FPs systematically using specific conditions and approaches resembling those of actual FFS experiments. Following this logic, we used typical FFS approaches (i.e., N&B and sFCS) to quantify the stability, brightness, pH sensitivity and *pf* of mEGFP, mNG, Gamillus and mNG.

A first unexpected result is that, in general, we do not observe strikingly different brightness values between each FP. While this observation can likely be explained by the distinctive experimental setup (e.g. very low excitation power, longer acquisition times, specific intra-cellular localization [21, 42, 44]), it is nevertheless relevant for users planning similar FFS-based investigations. Furthermore, the brightness measured via FFS is an average value derived from a statistical analysis of single molecule fluctuations. Such analysis is different from bulk methods that measure the fluorescence emission in a whole cell as a readout of brightness. While remaining both valid, the different approaches might provide information about different aspects of the fluorescent system. Finally, in our experimental conditions, Gamillus displayed a higher brightness compared to the other FPs, at least when located at the PM or while using higher laser powers. This observation was independently confirmed via both sFCS and N&B analysis.

Both techniques also indicate that the higher brightness of Gamillus is counterbalanced by a stronger tendency to photobleach, at least in the conditions used in our experiments (e.g., pH 7.4). A similarly low photostability was observed also for mGL, while mNG and mEGFP proved to be remarkably stable under continuous irradiation. Of interest, the contrast with reports indicating a strong photostability of mGL and Gamillus might be only apparent. A general comparison of these observations with previous studies is complicated by the fact that, often, different conditions and experimental setups result in different apparent photochemical behavior. For example, mEGFP was shown to be less stable than mNG under widefield illumination, more stable under laser illumination [23] and similarly stable using a spinning disk confocal microscope [42]. Nevertheless, the photostability observed here at pH 7.4 for mEGFP is in the same range as previously reported [23, 45]. Also, mGL was reported to be less

photostable than Clover that, in turn, is less photostable than mEGFP [21, 45], in agreement with our observations. On the other hand, in contrast to our observation, Shinoda *et al.* showed that Gamillus has a 2-fold higher photostability than mEGFP under widefield illumination and acidic conditions [24]. Once more, a possible explanation for such discrepancies might reside in the considerably different experimental conditions which, in this case, might be specifically relevant for FFS-based multimerization studies.

Next, we have measured the *pf* values for the different FPs, since this is a parameter of fundamental importance for quantitative multimerization studies. Our results indicate that, in general, all the examined FPs perform relatively well (i.e., *pf* ca. 0.7) and definitely better than most red or blue FPs [7, 19]. In particular, mGL excels in this context, with a *pf* value of 0.8 and above, as confirmed by both N&B and sFCS in different cell models. This might be due to its fast maturation rate [21].

In the second part of our work, we compared the FPs under different pH conditions. This is relevant for live-cell studies with FPs targeting e.g. acidic organelles, such as endosomes, secretory granules, lysosomes, and Golgi-Network (pH range ca. 4.7–8 [46]), which play a role for the sorting, transport and degradation of proteins [24, 25]. In general, mNG and mGL did not exhibit a special sensitivity to changes in pH values between 5.6 and 9.2. Neither of the monitored parameters (photostability, brightness or *pf*) displayed a significant correlation with pH values.

In contrast, mEGFP showed an increase of *pf* at acidic pH, with this value being significantly higher than that of Gamillus. We did not observe a strong decrease in mEGFP brightness at low pH values, as it was instead previously reported [26, 28, 43, 44]. This might be due to the low laser powers employed in our studies, since a control experiment performed at a ca. four-fold higher laser power resulted in a decrease in mEGFP brightness of ca. 40% (S7 Fig in S1 File). Apart from light-induced photochemical effect, it might also be possible that the data spread obtained at lower excitation power could partially mask a decrease in brightness. In any case, our data show that the brightness of mEGFP at pH 5.6 is indeed slightly lower than that of the other FPs.

Also, Gamillus was influenced by changes in pH value. First, we observed that its brightness remained higher than that of mEGFP, especially at low pH values. Interestingly, the photostability and the *pf* decreased dramatically as a function of decreasing pH values. In contrast with the idea of Gamillus being a FP particularly useful under acidic conditions [24], our data indicate that this fluorophore might perform best in basic environments in the context of FFS measurements. It is possible that strong differences between Gamillus and, e.g., mEGFP might be observed instead at pH values below those explored in this work (i.e. pH <5.5).

In conclusion, our results indicate that mEGFP is in general a very good choice for the quantification of multimerization via FFS, also at pH values between 5.6 and 7.4. Although its brightness is lower than other examined FPs, its remarkable photostability would allow using higher excitation powers (thus increasing its brightness). Bright FPs, such as Gamillus, could be efficiently used for e.g. qualitative imaging with short acquisition times and low excitation powers. At neutral pH conditions, a remarkably high *pf* was observed for mGL, although care should be taken in experiments in which photobleaching might represent an issue. Finally, at basic pH conditions (e.g., up to 9.2), Gamillus represents an optimal choice based on a good photostability, high *pf* and brightness.

## Supporting information

**S1 File. Supporting materials and methods, supporting tables and supporting figures.**
(DOCX)

**S1 Data.**
(XLSX)

## Acknowledgments

We thank all the members of the Physical Biochemistry group for useful feedback.

## Author Contributions

**Conceptualization:** Annett Petrich, Amit Koikkarah Aji, Valentin Dunsing, Salvatore Chiantia.

**Formal analysis:** Annett Petrich, Amit Koikkarah Aji, Salvatore Chiantia.

**Funding acquisition:** Salvatore Chiantia.

**Investigation:** Annett Petrich, Amit Koikkarah Aji.

**Project administration:** Salvatore Chiantia.

**Software:** Annett Petrich, Valentin Dunsing, Salvatore Chiantia.

**Supervision:** Salvatore Chiantia.

**Writing – original draft:** Annett Petrich.

**Writing – review & editing:** Amit Koikkarah Aji, Valentin Dunsing, Salvatore Chiantia.

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
