## [Decision Letter · Decision Letter 0]

12 Mar 2023

PONE-D-23-04438Benchmarking of novel green fluorescent proteins for the quantification of protein oligomerization in living cellsPLOS ONE

Dear Dr. Chiantia,

Thank you for submitting your manuscript to PLOS ONE. After careful consideration, we feel that it has merit but does not fully meet PLOS ONE’s publication criteria as it currently stands. Therefore, we invite you to submit a revised version of the manuscript that addresses the points raised during the review process.

We look forward to receiving your revised manuscript.

Kind regards,

Rajiv Kumar Kar, Ph.D.

Academic Editor

PLOS ONE

“We thank all the members of the Physical Biochemistry group for useful feedback. VD is grateful for

373 support by an HFSP long-term postdoctoral fellowship (HFSP LT0058/2022-L). This work was partially

374 supported by the Deutsche Forschungsgemeinschaft (DFG) project number 407961559 to S.C”

“Deutsche Forschungsgemeinschaft (DFG) project number 407961559 to S.C.

www.dfg.de

HFSP long-term postdoctoral fellowship (HFSP LT0058/2022-L) to V.D.

www.hsfp.org

4.Please review your reference list to ensure that it is complete and correct. If you have cited papers that have been retracted, please include the rationale for doing so in the manuscript text, or remove these references and replace them with relevant current references. Any changes to the reference list should be mentioned in the rebuttal letter that accompanies your revised manuscript. If you need to cite a retracted article, indicate the article’s retracted status in the References list and also include a citation and full reference for the retraction notice.

Additional Editor Comments:

The authors are furthermore requested to submit the raw data of their experimental results in the supporting information. Authors must ensure the reproducibility of the results by providing explicit details of the methodologies (and protocols) involved in this work.

Reviewers' comments:

Reviewer's Responses to Questions

**Comments to the Author**

1. Is the manuscript technically sound, and do the data support the conclusions?

Reviewer #1: Yes

Reviewer #2: Yes

2. Has the statistical analysis been performed appropriately and rigorously? 

Reviewer #1: I Don't Know

Reviewer #2: Yes

3. Have the authors made all data underlying the findings in their manuscript fully available?

Reviewer #1: Yes

Reviewer #2: Yes

4. Is the manuscript presented in an intelligible fashion and written in standard English?

Reviewer #1: Yes

Reviewer #2: Yes

5. Review Comments to the Author

Reviewer #1: Petrich et al. compared the performance of several green fluorescent proteins in live cell imaging and tried to establish a benchmark for selecting appropriate tags for quantitative microscopy. The experiments were well-designed and supported by statistical analysis. The manuscript is well-written and suitable for PlosOne. However, it requires minor revisions to fix the following issues.

1. What is the basis for selecting the linker sequences? What would be the result if the linker sequence were changed?

2. In Figure 2A, to explain brightness, the authors used interquartile range (IQR), which has a scattered value compared to the mean of all readings. However, the authors did not explain the significance of IQR.

3. The authors are recommended to mention the goodness of fit in Fig. S1D and Fig. S6. In

4. Figure S6, the authors are recommended to improve the fitting as, in certain cases, the error is more than 20%.

5. To follow the benchmark set by the authors, they should mention the curve fitting equations used in each analysis.

6. The authors are recommended to provide a tabulated summary for the use of different FP tags based on the results of the current study. This will make it easier for readers to follow the outcomes of this study and apply them in their work.

Reviewer #2: This manuscript is about the luminescent properties of some novel GFP variants. The authors have carried out the ‘number and brightness’ analysis as well as the scanning FCS. They have previously published similar work with other fluorescent proteins, keeping the same application (oligomerization) in the mind (e.g. Sci Rep. 2018; 8: 10634). This study, therefore, is an incremental contribution to the already published reports, and could be of use for researchers employing GFP variants for studying protein oligomerization. The study may be considered for publications provided the following minor issues are addressed:

1. As it is a follow up study, the authors have to be careful about the plagiarised text. The sentence spanning lines 156-159 “To quantify the amount of non-fluorescent FPs, we consider all these processes together in a single parameter, the apparent fluorescence probability (pf), i.e. the probability of a FP to emit a fluorescence signal”, for example, is an identical copy from their previous work in Scientific Reports (2018). There could be other similar issues that might have been overlooked by the authors.

2. Abstract: “several cellular functions”. Several is used to indicate more than two but not too many. I think “many cellular functions” will fit better in this sentence.

3. The abbreviation FB is for ‘fluorescent protein’, and not ‘fluorescence protein’, I guess.

4. “provided Fig S1A” should be “provided in Fig. S1A”.

5. There are several typographical errors that the authors need to carefully check and correct.

6. PLOS authors have the option to publish the peer review history of their article (what does this mean?). If published, this will include your full peer review and any attached files.

Reviewer #1: No

Reviewer #2: No

---

## [Author Response · Author response to Decision Letter 0]

5 Apr 2023

We have taken into account both documents now.

[…]

[…]

We have corrected the Acknowledgments section. The Funding Statement is correct as is.

We have now explicitly added the data as S7 Fig (line 362)

The reference list is complete and correct.

Additional Editor Comments:

The authors are furthermore requested to submit the raw data of their experimental results in the supporting information. Authors must ensure the reproducibility of the results by providing explicit details of the methodologies (and protocols) involved in this work.

The raw data related to the manuscript figures is now included in the submission. We have also included more details in the Methods section.

 

Reviewer #1: Petrich et al. compared the performance of several green fluorescent proteins in live cell imaging and tried to establish a benchmark for selecting appropriate tags for quantitative microscopy. The experiments were well-designed and supported by statistical analysis. The manuscript is well-written and suitable for PlosOne. However, it requires minor revisions to fix the following issues.

1. What is the basis for selecting the linker sequences? What would be the result if the linker sequence were changed?

This is a very good point: in the first submission, we have listed the precise sequence of linkers in homo-dimers constructs (Table S1) but we have not explicitly indicated whether there was a rational for the choice. 

Previously, we have shown that already a flexible 7 aa linker is enough to avoid significant energy transfer in a homo-oligomer and artifacts in brightness measurements (Dunsing et al, 2018, Sci. Rep.). The energy transfer efficiency might be even lower for linkers without glycines (van Rosmalen et al., 2017, Biochemistry; Arai et al, 2001, Design and Selection). Therefore, we simply used long (i.e., ≥ 7 aa) linkers for the constructs in this study. For constructs which were already available to the public (e.g., the mEGFP constructs already available via Addgene), we did not change such long flexible linkers. For new constructs, we used 7-10 aa long rigid linkers (i.e. prolin-rich linkers containing the sequence PPAAAPP), with the exception of mNG. In this case, we had to use a 7 aa long flexible linker, due to difficulties in the cloning procedure caused by some restriction sites being within the protein sequence.

Finally, it is worth noting that Wu et al. demonstrated that the brightness of FP homo-dimers is not significantly affected by homo-FRET (Wu et al, 2009, BiophysJ). For this reason, we expect that the specific choice of linkers is not particularly determinant for the measurements presented here. 

We added a comment at this regard in the SI, together with the corresponding table S1.

2. In Figure 2A, to explain brightness, the authors used interquartile range (IQR), which has a scattered value compared to the mean of all readings. However, the authors did not explain the significance of IQR.

All the results regarding brightness (including discussion and statistical tests) are strictly based on the observed mean values. The main text has been now modified accordingly (lines 222-3. All line numbers refer to the document with Track Changes). The IQR is reported exclusively for a matter of clarity and completeness (as e.g., for the median in Fig. 2A or the SEM in Fig. 2B) and refers simply to the distance between horizontal lines in the box plots. Apart from showing the general spread of the data, we do not attribute further relevant meaning to the parameter. 

3. The authors are recommended to mention the goodness of fit in Fig. S1D and Fig. S6. 

AND

4. In Figure S6, the authors are recommended to improve the fitting as, in certain cases, the error is more than 20%.

Since Fig. S1D is simply showing the experimental workflow and contains no relevant data, we will focus here on Fig. S6. This figure shows typical autocorrelation curves and the fitting according to a 2d diffusion model. The level of noise (e.g., in correspondence of short time lags and emphasized by the semi-logarithmic scale) is, in general, comparable to what can be expected from similar measurements in living cells (see e.g. Sigaut et al., 2021, PONE; James et al., 2021, Appl. Sci.), also according to our experience. Most importantly, in the case of the measurements shown specifically in this work, it must be noted that generally low, sub-optimal laser powers had to be used. On one hand, this was needed to be able to compare different fluorescent proteins with different photo-stabilities. On the other hand, sub-optimal laser powers will naturally decrease the signal-to-noise ratio, as noticed by the reviewer. This compromise is nevertheless necessary in such a comparison study. We have added a comment at this regard in caption of Fig. S6.

Regarding the issue of possibilities to improve the fitting, we would prefer not to change the fitting procedure used here, since the same model has been consistently used by us and other groups for similar experiments (e.g., Ries et al., 2008, Proceedings of SPIE; Schneider et al., 2018, ACS Nano; Petrich et al., 2021, BiophysJ). Although the uncertainty on the final parameters can be relatively high (due to the low signal-to-noise ratio, as explained in the previous paragraph), there is no recognizable systematic deviation in the fit residues (i.e., consistently present for all samples) that would justify changing the classical 2d diffusion model for a model with e.g. more degrees of freedom.

5. To follow the benchmark set by the authors, they should mention the curve fitting equations used in each analysis.

We have added further explanations regarding the fitting models used in the methods section (lines 157-160) and the captions to Fig. S1 and Fig. S6.

6. The authors are recommended to provide a tabulated summary for the use of different FP tags based on the results of the current study. This will make it easier for readers to follow the outcomes of this study and apply them in their work.

All the results are summarized in tables Tab S2 and the new tables Tabs S3 and S4 (the latter ones containing the relevant pf information).

 

Reviewer #2: 

1. As it is a follow up study, the authors have to be careful about the plagiarised text. The sentence spanning lines 156-159 “To quantify the amount of non-fluorescent FPs, we consider all these processes together in a single parameter, the apparent fluorescence probability (pf), i.e. the probability of a FP to emit a fluorescence signal”, for example, is an identical copy from their previous work in Scientific Reports (2018). There could be other similar issues that might have been overlooked by the authors.

We thank the reviewer for the careful reading. Indeed, some sentences (especially in the Methods section) were very similar to previous publications. This is understandable, since those passages deal with technical details and definitions which do not change between different experiments, but it should be nevertheless avoided. For this reason, we have changed the wording in many passages, especially in the Methods section. The content stayed, of course, mostly unchanged. See e.g. lines 49-51, 96-100, …, 306-310. All line numbers refer to the document with Track Changes. 

2. Abstract: “several cellular functions”. Several is used to indicate more than two but not too many. I think “many cellular functions” will fit better in this sentence.

Amended.

3. The abbreviation FB is for ‘fluorescent protein’, and not ‘fluorescence protein’, I guess.

Amended.

4. “provided Fig S1A” should be “provided in Fig. S1A”.

Amended.

5. There are several typographical errors that the authors need to carefully check and correct.

We have once more read the whole manuscript and corrected few typos.

---

## [Decision Letter · Decision Letter 1]

25 Apr 2023

Benchmarking of novel green fluorescent proteins for the quantification of protein oligomerization in living cells

PONE-D-23-04438R1

Dear Dr. Chiantia,

We’re pleased to inform you that your article has been adequately amended and has now been accepted for publication. However, it must meet all outstanding technical requirements.

Best wishes and thank you for choosing PLOS One to publish your work.

Within one week, you’ll receive an e-mail detailing any required amendments. When these have been addressed, you’ll be notified and your manuscript will be scheduled for publication.

Kind regards,

Rajiv Kumar Kar, Ph.D.

Academic Editor

PLOS ONE

Reviewers' comments:

Reviewer's Responses to Questions

**Comments to the Author**

1. If the authors have adequately addressed your comments raised in a previous round of review and you feel that this manuscript is now acceptable for publication, you may indicate that here to bypass the “Comments to the Author” section, enter your conflict of interest statement in the “Confidential to Editor” section, and submit your "Accept" recommendation.

Reviewer #1: All comments have been addressed

Reviewer #2: All comments have been addressed

2. Is the manuscript technically sound, and do the data support the conclusions?

Reviewer #1: Yes

Reviewer #2: Yes

3. Has the statistical analysis been performed appropriately and rigorously? 

Reviewer #1: Yes

Reviewer #2: Yes

4. Have the authors made all data underlying the findings in their manuscript fully available?

Reviewer #1: Yes

Reviewer #2: Yes

5. Is the manuscript presented in an intelligible fashion and written in standard English?

Reviewer #1: Yes

Reviewer #2: Yes

6. Review Comments to the Author

Reviewer #1: Petrich et al. compared the performance of several green fluorescent

proteins in live cell imaging and tried to establish a benchmark for selecting appropriate

tags for quantitative microscopy. The experiments were well-designed and supported

by statistical analysis. The manuscript is well-written and suitable for PlosOne.

They have fixed the issues and answered the queries convincingly.

Reviewer #2: (No Response)

7. PLOS authors have the option to publish the peer review history of their article (what does this mean?). If published, this will include your full peer review and any attached files.

Reviewer #1: No

Reviewer #2: No

---

## [Editor Report · Acceptance letter]

28 Apr 2023

PONE-D-23-04438R1 

Benchmarking of novel green fluorescent proteins for the quantification of protein oligomerization in living cells 

Dear Dr. Chiantia:

I'm pleased to inform you that your manuscript has been deemed suitable for publication in PLOS ONE. Congratulations! Your manuscript is now with our production department. 

Kind regards, 

on behalf of

Dr. Rajiv Kumar Kar 

Academic Editor

PLOS ONE